# Study of the Relationship between the Average Annual Temperature of Atmospheric Air and the Number of Tick-Bitten Humans in the North of European Russia

**DOI:** 10.3390/ijerph17218006

**Published:** 2020-10-30

**Authors:** Andrei Tronin, Nikolay Tokarevich, Olga Blinova, Bogdan Gnativ, Roman Buzinov, Olga Sokolova, Birgitta Evengard, Tatyana Pahomova, Liliya Bubnova, Olga Safonova

**Affiliations:** 1Scientific Research Centre for Ecological Safety of the Russian Academy of Sciences, St. Petersburg Federal Research Center of the Russian Academy of Sciences, 18, Korpusnaya str., 197110 St.-Petersburg, Russia; 2Saint-Petersburg Pasteur Institute, 14, str. Mira, 197101 St.-Petersburg, Russia; zoonoses@mail.ru (N.T.); aerolga@yandex.ru (O.B.); 3Komi Republic Office of the Russian Federal Service for Surveillance on Consumer Rights Protection and Human Wellbeing, 71, Ordjonikidze str., 167016 Syktyvkar, Republic of Komi, Russia; gnativ_br@mail.ru; 4Department of hygiene and medical ecology, The Northern State Medical University, 51, Troitskiy Ave., 163000 Arkhangelsk, Arkhangelskaya oblast, Russia; arkh@29.rospotrebnadzor.ru (R.B.); sokolovaov@29.rospotrebnadzor.ru (O.S.); 5Arkhangelsk Regional Office of the Russian Federal Service for Surveillance on Consumer Rights Protection and Human Wellbeing, 24, Gaydar str., 163000 Arkhangelsk, Arkhangelskaya oblast, Russia; 6Dept. Clinical Microbiology, Umea University, 90187 Umea, Sweden; birgitta.evengard@umu.se; 7Karelia Republic Regional Office of the Russian Federal Service for Surveillance on Consumer Rights Protection and Human Wellbeing, 12, Pirogov str., 185002 Petrozavodsk, Republic of Karelia, Russia; pahomova@cge.onego.ru (T.P.); bubnova@cge.onego.ru (L.B.); safonova@cge.onego.ru (O.S.)

**Keywords:** climatic changes, annual air temperature, tick-bitten humans

## Abstract

In recent decades, a considerable increase in the number of tick-bitten humans has been recorded in the north of European Russia. At the same time, significant climatic changes, such as an increase in air temperature, were noticed in this region. The northern border of the *ixodidae* distribution area lies in the north of European Russia, therefore the analysis of the population dynamics is of particular interest regarding the possible impact of the climate changes. Unfortunately, in such a large territory field, studies on tick abundance are very difficult. In our study, the official statistics for the number of tick-bitten humans were used. This kind of statistical analysis has been conducted in the Russian Federation for many years, and can be used for the estimation of climate change impact on tick abundance. Statistical data on tick-bitten humans have been collected in three large regions for several decades. For the same regions, the average annual air temperature was calculated and modeled. An S-shaped distribution of the number of victims depending on the average annual air temperature was established, which can be described as “Verhulst’s law”, or logistic function. However, the development of the population does not depend on time, but on the temperature of the ambient air.

## 1. Introduction

*Ixodidae* ticks are known to be vectors of many pathogens that cause dangerous infectious diseases in humans, such as tick-borne encephalitis and Lyme borreliosis, human monocytic ehrlichiosis, human granulocytic anaplasmosis and others. Therefore, a quantitative estimate of tick population is necessary when developing preventive measures within any administrative territory. Usually the classic method of flagging, collection from a sheet, is applied [1,2,3]. However, the results for separate individual areas where ticks were collected do not represent in a relevant manner the abundance of *ixodidae* over very large areas, such as the north of European Russia. Moreover, the monitoring of tick abundance in Russia is very fragmentary [1,2,3].

In our study, the official statistics of the number of tick-bitten humans were used. This method was already used by other researchers [4,5]. Despite significant limitations (see Section 4 below), such an approach allows the identifying of “novel” tick habitats, where the flagging method is not efficient due to the small size of the tick population, while the state healthcare system adequately registers all complaints of tick-bitten humans. In addition, long-period tick-bite incidence rate (TBIR) monitoring makes it possible to register the time dependence of tick abundance over the large territory.

Numerous studies on the impact of climate change on the ixodid tick’s distribution have been carried out in Russia, Europe and North America [6,7,8,9,10]. The range of ticks in European Russia has been shown to spread northward, the air temperature being the main driver of this effect [11,12]. Simulation of the tick population’s development upon predicted climatic changes was carried out for the entire territory of Russia [13]. A separate line of research is represented by simulation of the development of the ixodid tick population with time [14,15,16].

Previously, it was suggested that in the Komi Republic, the dependence of TBIR on the average annual air temperature can be approximated by the “Malthus equation”, that is, by an exponential law. At the same time, a high level of correlation of 0.84 (*p* < 0.0001) was shown between TBIR and the mean annual air temperature in the Komi Republic. Further, it was hypothesized that the exponential growth of TBIR does not continue indefinitely, so the “Verhulst equation” is more consistent with the real picture than the “Malthus equation” [17].

Here, an attempt is made to study the impact of air temperature variation on the dynamics of the ixodid tick population in the north of European Russia, relying on TBIR data, and to test the hypothesis about the possibility of its simulation using the logistic function (“Verhulst’s law”), if the area under study involves the neighboring territories (Arkhangelsk Oblast and the Republic of Karelia), their climate and environment (northern taiga) being similar to those of Komi Republic (Figure 1).

## 2. Materials and Methods

### 2.1. Epidemiological Data

All registered cases seeking medical help after tick bites in the Arkhangelsk Oblast area in 1980–2016 (19 administrative units), in the Komi Republic in 1992–2014 (19 administrative units) and in the Republic of Karelia in 2002–2016 (17 administrative units) were analyzed. For many years, the population has been encouraged to seek medical assistance when experiencing a tick-bite, and this is common procedure in Russia. Since the habitancy varied during the study period, for comparison purposes the tick-bite incidence rate (TBIR) was used, which is the number of humans bitten by ticks during the year, per 100 thousand of the local population.

The primary information (administration unit and date of the bite) was provided by treatment and prevention organizations in the “Centers of Hygiene and Epidemiology” of those three regions. TBIR was calculated by taking into account the official data on the human population of each territory in the relevant year.

### 2.2. Environmental Data

The studied territory is located in the north of European Russia (Figure 1). It is mostly taiga, split into northern (~50%) and middle (~50%) subzones. Only the northernmost areas of the Arkhangelsk Oblast and the Komi Republic belong to the forest-tundra zone. There are many swamps, and natural meadows along river valleys. In taiga, the main tree species are spruce, fir, larch and pine, while birch and alder occur as well. The herb-shrub layer includes lingonberry, bilberry and blueberry, and various herbs. In taiga, as a rule, a moss cover of boreal green mosses is well developed. The fauna are represented by elk, deer, brown bear, wolf, fox, squirrel, beaver and hare, and various gnawing animals are widespread. A wide set of migratory and sedentary bird species are very common, especially in the north part of research area—from the Golden eagle to the Little bunting, and from the Whooper swan to the Eurasian bullfinch. A few amphibian species, such as the common frog, Moor frog and common toad, could be found in this area.

The climate of the area is cold continental, subarctic, with long winters (mild in the west but harsh in the east), short cool summers and permanent moisture throughout the year. Average annual air temperatures range from negative in the north of the Komi Republic to +6 °C in the south of Karelia. The territory belongs to the zone of excessive moisture, as the average annual amount of atmospheric precipitation is 400–750 mm, and it increases southward. In the second half of the 20th century, significant climate changes, especially in terms of air temperature, were observed in the north of European Russia [18].

### 2.3. Meteorological Data

Meteorological data (air temperature in 1948–2016) were provided by the National Centers for Environmental Information, USA [19]. The “CPC Monthly Global Surface Air Temperature Data Set” represents the average monthly air temperature at a height of 2 m, gridded with a certain step. The grid step is 0.5 degrees for the entire globe, and the time interval is from 1948 to the current moment. Gridded air temperatures were recalculated using ArcGIS software (version 10.2, ESRI, Redlands, CA, USA) to the temperatures of the administrative districts of the Arkhangelsk Oblast, the Republic of Karelia and the Republic of Komi. Average annual air temperature, *AT_j_*, was calculated from monthly average data using the formula:(1)$$AT_{j}\;=\;\frac{1}{12}\;\sum_{i=1}^{12}\left(T_{ij}\right)$$
Here, *T_ij_* is the monthly average air temperature in “*i*” month, at “*j*” administrative unit (a district of Komi, Karelia or Arkhangelsk Oblast).

In this manner, the average annual air temperature, *AT_j_*, was calculated for all administrative districts within the area under study. The average annual air temperature is subject to strong fluctuations from year to year. To isolate climatic changes, various mathematical procedures are applied to the original data on air temperature (averaging, trend selection, etc.). To avoid interannual fluctuations of the average annual air temperature, a second-order polynomial function was applied to the average annual air temperature of all administrative units of the region studied with the help of Microsoft Excel’s (Microsoft Office 2019, Microsoft Corporation, Redmond, WA, USA) interpolation tools.

The model temperature is calculated by the formula:*mAT_j_* = *a* × *AT_j_*^2^ + *b* × *AT_j_* + *c*(2)
Here, *mAT_j_* is the average annual model air temperature (°C), *AT_j_* is the average annual observed air temperature (°C), and *a*, *b* and *c* are the calculated coefficients.

Using a polynomial function, average annual model air temperatures (*mAT_j_*) for 1948–2016 were calculated for each administrative district of the Arkhangelsk Oblast, Karelia and Komi. Hereinafter, all references to air temperature should be taken as the average annual model air temperature.

The *mAT_j_* and TBIR values were used for all administrative units of the Arkhangelsk Oblast, Karelia and Komi, brought together in a database. Each record in the database involves the name of the administrative unit, the year of the event, *mAT_j_* and TBIR. The database contains 1395 records, multiplying the number of years of *mAT_j_* and TBIR study by the number of administrative units, resulting in 703 for the Arkhangelsk Oblast, 255 for the Republic of Karelia, and 437 for the Komi Republic.

## 3. Results

All registered cases seeking medical help after tick-bites in the Arkhangelsk Oblast, the Komi Republic and in the Republic of Karelia were considered. In total, 187,722 tick bites have been registered over the period under study—102,791 in the Arkhangelsk Oblast, 18,690 in the Komi Republic and 66,241 in the Republic of Karelia. Figure 2 represents tick-bite incidence rate (TBIR) in the Arkhangelsk Oblast, and the Karelia and Komi republics.

### 3.1. Relationship between the Average Annual Temperature of Atmospheric Air and the Number of Tick-Bitten Humans

To study the relationship between *mAT_j_* and TBIR, a two-dimensional diagram was constructed, where the abscissa is *mAT_j_* and the ordinate is TBIR on a logarithmic scale (see Figure 3). To plot a chart on a logarithmic scale to base 10 for the TBIR values, “0” in our sample was replaced with “1”. This did not disturb our results, since the TBIR in the study areas usually exceeded 1. Since the population of those administrative units rarely exceeded 100,000, this means one tick bite per year. This is a very small value, virtually close to zero.

It is much more convenient to analyze samples in this form. In our opinion, this statistical population involves at least two groups, as follows:The first group of samples correspond to districts with significant numbers of recorded tick-bites, and, accordingly, those places provide favorable habitats for ixodid ticks. The local average annual air temperature is usually above zero;The second group includes the districts where tick bites are not recorded. The local temperature varies between −7 °C and +3 °C.

Of greatest interest is the first group, which provides information about the dependence of the number of tick bites on the temperature. The second group consists of districts where there are practically no tick-bites. Only a few tick bites on record are, as a rule, imported cases.

For a more detailed investigation of the first group, the second group was deleted by removing all points corresponding to TBIRs between 0 and 1, i.e., negligibly small values (see Figure 4).

Our analysis of the air temperature dependence of nonzero TBIRs has revealed three TBIR zones (Figure 4): the invasion zone (blue color), the exponential growth zone (red) and the saturation zone (green). The invasion zone is characterized by low air temperatures from −5 °C to +0.5 °C, and relatively low TBIR (less than 100). The zone corresponds to tick penetration into the district. The zone is interesting because it shows the possibility of tick bites in places with very low air temperatures. It cannot be ruled out that ticks are brought to these territories by migratory birds, but the low air temperatures do not provide the conditions satisfactory for their survival [1,2,3]. The zone of exponential growth is characterized by TBIR increase with temperature. The temperature increases from +0.5 °C to +4 °C, while for TBIR an explosive growth from 100 to several thousand is noted. Next follows a saturation zone, where the number of tick-bitten humans remains unchanged independently of the temperature growth. Within this zone, the air temperature varies from +4 °C to +6 °C, while the TBIR fluctuates around 1000. The saturation zone is remarkable for the fact that at temperatures above +4 °C, the local TBIR value is about 1000, and practically never falls below. This means that at temperatures above +4 °C, saturation of the ixodid ticks population occurs, and all northern taiga districts of European Russia have got TBIR values of about 1000.

The analysis of TBIR distribution as a function of air temperature shows that it does not follow the exponential “Malthus equation”, as indicated by its relatively low correlation coefficient of 0.67 (*p* < 0.0001).

### 3.2. Application of the “Verhulst Equation” to Describe the Air Temperature Dependence of TBIR

As mentioned earlier, the sample offered for consideration can be described by the “Verhulst equation”:(3)$$\mathrm{TBIR}=\frac{K*\mathrm{TBIR}0\times e^{r\times mAT_{j}}}{K+\mathrm{TBIR}0\times (e^{r\times mAT_{j}}-1)}$$
Here, TBIR is the number of tick-bitten humans per 100 thousand of the population, TBIR0 is the initial number of tick bites per 100 thousand of the population, *K* is the carrying capacity (maximum possible size of a population), *r* is the growth rate, and *mAT_j_* is the modeled air temperature.

To calculate those coefficients (*r* and TBIR0), the distribution for the Arkhangelsk Oblast was used, as the most representative distribution in the area of exponential growth. Based on the distribution (Figure 4), we took *K* to be 1000. We added one to the calculated value for its correct representation on the graph. The graph of the “Verhulst equation” is shown in Figure 5.

In our opinion, the “Verhulst equation” quite adequately describes the dependence of the TBIR on air temperature in the north of European Russia, because it reflects all three areas of distribution: the area of invasion, the area of exponential growth, and the area of saturation.

### 3.3. Dependence of TBIR on Air Temperature by Region and over Time

The dependence of TBIR on temperature in three regions (Arkhangelsk Oblast, Republic of Karelia and the Komi Republic) is presented in Figure 6.

Several conclusions can be drawn from the analysis of the TBIR distribution’s dependence on the air temperature in the regions. The above-mentioned zone of invasion is associated with the Komi Republic, where the air temperature is the lowest. Arkhangelsk Oblast is within the zone of exponential growth. The Republic of Karelia forms both the zone of exponential growth and that of saturation. We can observe a shift in the distribution in time from the region of low temperatures (Republic of Komi) to the region of high temperatures (Republic of Karelia). The population of ticks in the zone of saturation in the Republic of Karelia was formed in the 1990s and early 2000s [20], while in the Komi Republic it is being actively formed now [12].

Comparison of TBIRs in the analyzed territories over time (Figure 2) allows us to state that their tendencies are opposing. Thus, in the Arkhangelsk Oblast and the Komi Republic, there is a pronounced increase in the TBIR, while in Karelia the TBIR has been noticeably decreasing since 2003. Upon detailed examination of those trends, one may note that in recent years, since 2011, the TBIR in Arkhangelsk Oblast stabilized, while in Karelia it slightly decreased.

## 4. Discussion

Over recent decades, the biotic components of landscapes changed significantly in the north of European Russia. The northward shift of the forest borders has led to the dispersal of many species of wild mammals which are the principal hosts of ixodid ticks. Previously, the northern border of the ixodid tick’s habitat was much further south [21]. Over a forty-year observation period, the northward shift of the tick distribution limit in the Komi Republic was at least 150–200 km [22]. Similar processes took place in the Arkhangelsk Oblast [23] and the Republic of Karelia [24].

In our study, to estimate the dynamics of the tick population, long-term statistics provided by Rospotrebnadzor institutions, which recorded all medical care encounters in connection with tick-bites, were used. The approach has certain advantages over the classical method of collecting with flags or traps. The analysis of the long-term tick-bite incidence rate (TBIR) data allows the indirect assessing of the tick abundance dynamics on a large territory (not only within the sites of collection), and the estimating of the related epidemiological hazard, since it directly quantifies tick attacks on humans.

On the other hand, the method has got some limitations, since its result depends on the availability of medical care, the awareness of the population about the danger of ticks, and the social conditions that determine the frequency of human exposure to ticks. Although those factors may influence the analysis, the really significant increase in TBIR in the Arkhangelsk Oblast and the Komi Republic was previously confirmed by a sharp rise in the incidence of tick-borne encephalitis (TBE) [11].

TBE cases were recorded, inter alia, in the northern districts, where the presence of ticks was attested only due to the health-seeking behavior of tick-bitten humans. Moreover, our study shows that in recent years, TBIR decreased in the Republic of Karelia. This trend can hardly be explained by a loss of awareness or a decrease in the availability of medical care.

The social conditions of life in the Republic of Karelia do not reduce, but rather increase the risk of exposure to tick bites: the woodworking industry is on the rise, territories for summer cottages are being developed, and the number of private cars increases, thus making the local population more mobile. This justifies, in our opinion, the applicability of the long-term TBIR data analysis to the indirect estimation of the dynamics of tick abundance over large territories of Russia.

The analysis of the TBIR’s temperature dependence shows the ambient air temperature to be the main, though not the sole, driver of ixodid tick abundance in the north of European Russia [1,2,3]. In approximately similar ecosystems of the northern taiga and under excessive moisture, it is temperature that plays the main role in the formation of the tick population [2,3].

Studies of the *I. persulcatus* tick’s ecology have shown that the development of the tick population requires the accumulated sum of air temperatures to be either 1400–1500 °C over the period with stable average daily temperatures exceeding 10 °C, or 1600 °C over the period with a stable average daily temperature exceeding 5 °C [2,3]. The same authors emphasized the importance of temperature conditions for the population wintering, those being determined by the air temperature and the characteristics of the snow cover. In our opinion, under the conditions of the north of European Russia, it is advisable to use the average annual temperatures as reflecting the region’s temperature regime in total.

The revealed variations in the TBIR trends suggest that, despite the long-term uptrend in the air temperature over the north of European Russia [16], the increase in tick abundance, even under favorable temperature conditions and sufficient precipitation, is not endless. Since approximately 2008, both in the Republic of Karelia and in the Arkhangelsk Oblast, the TBIR has remained at the same level, and ranges from 400 to 600 (Figure 2).

## 5. Conclusions

The S-shaped distribution of the tick-bite incidence rate (TBIR) from the mean annual air temperature in the north of the European territory of Russia is established, whereby the TBIR is presented in a logarithmic form. The TBIR’s dependence on the air temperature can be described by “Verhulst’s law” or a logistic function, but the population development depends not on time, but on the ambient air temperature.

The first tick-bites on humans were recorded in truly Arctic regions, with local average annual air temperatures close to zero. The exponential growth of the TBIR is observed in areas with temperatures from +0.5 °C to + 4 °C. The “saturation” conditions, i.e., the TBIR’s growth’s arrest in spite of air temperature growth, are observed at average annual air temperatures exceeding +4 °C.

A TBIR value of about 1000 may be considered as its saturation value. That is, the TBIR grows with the regional air temperature (due to climatic changes) to 1000, sometimes exceeds it, and then tends back to the level of 1000 and does not grow further. Therefore, a preliminary conclusion can be drawn that the size of tick population also remains stable, and does not grow further with increasing temperature.

## Figures and Tables

**Figure 1 ijerph-17-08006-f001:**
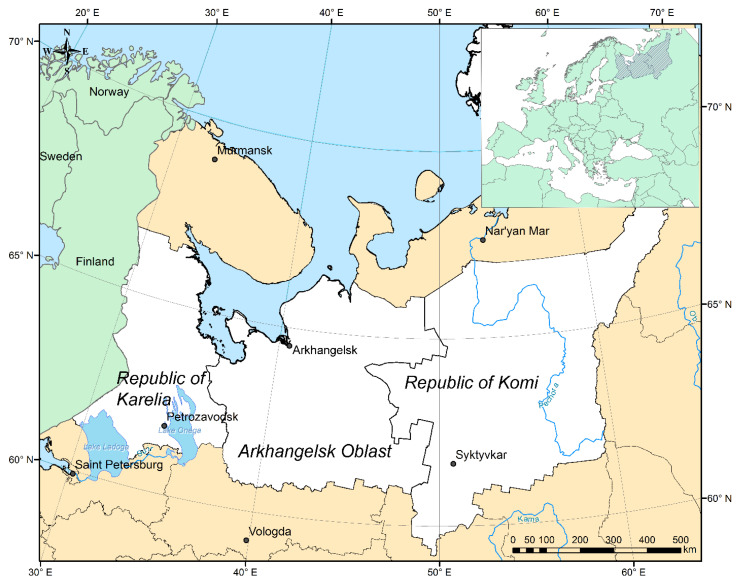
Geographical location of the Arkhangelsk region, the Republic of Karelia and the Komi Republic in the north of European Russia.

**Figure 2 ijerph-17-08006-f002:**
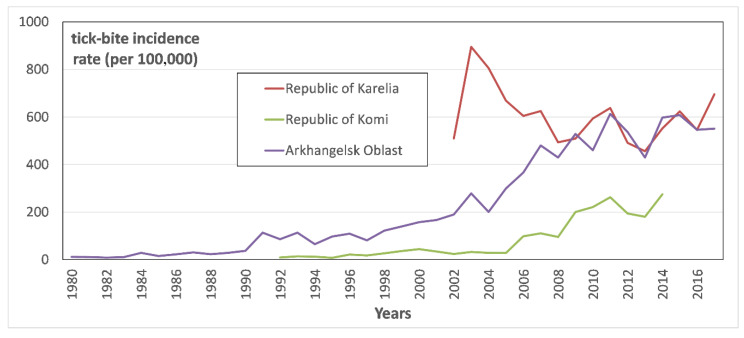
TBIR in the Arkhangelsk Oblast, the Republics of Komi and Karelia in 1980–2016.

**Figure 3 ijerph-17-08006-f003:**
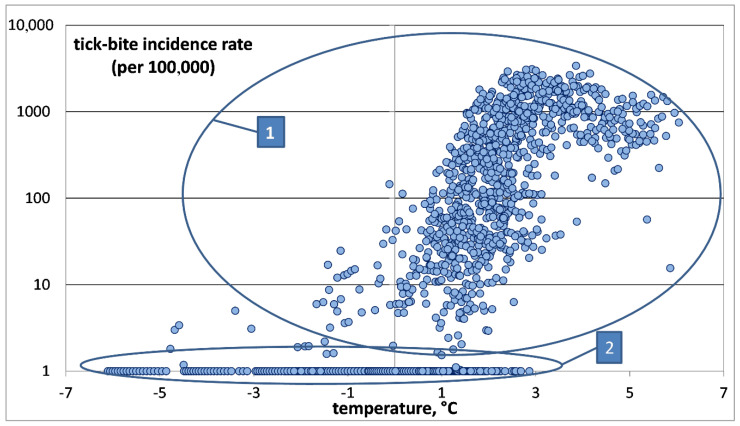
Tick-bite incidence rate (ordinate) versus average annual model air temperature (abscissa) for all administrative units of the Arkhangelsk Oblast, Karelia and Komi. TBIR on a logarithmic scale. Separate sample groups are highlighted in the figure. The description is given in the text.

**Figure 4 ijerph-17-08006-f004:**
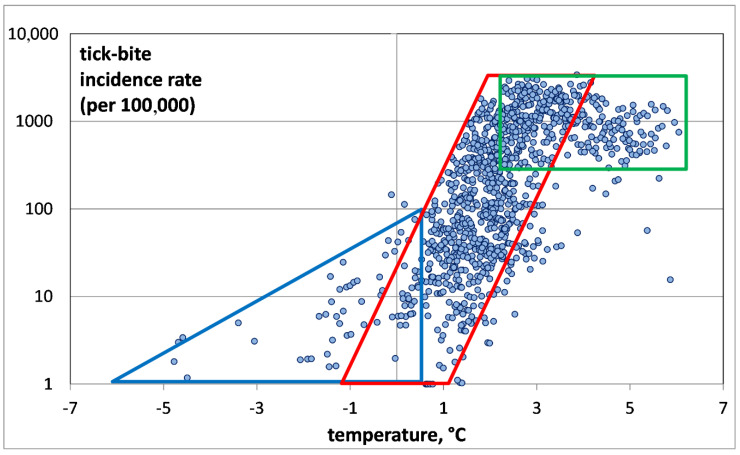
Nonzero tick-bite incidence rate versus average annual model air temperature for all administrative units of the Arkhangelsk Oblast, Karelia and Komi. The highlighted areas are represented as the invasion zone (blue color), the exponential growth zone (red) and the saturation zone (green).

**Figure 5 ijerph-17-08006-f005:**
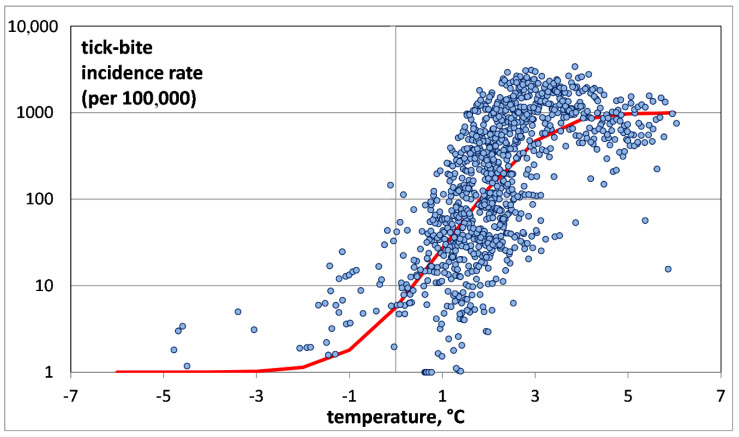
Tick-bite incidence rate versus average annual model air temperature for all administrative units of the Arkhangelsk Oblast, Karelia and Komi, and the “Verhulst equation” (shown in red).

**Figure 6 ijerph-17-08006-f006:**
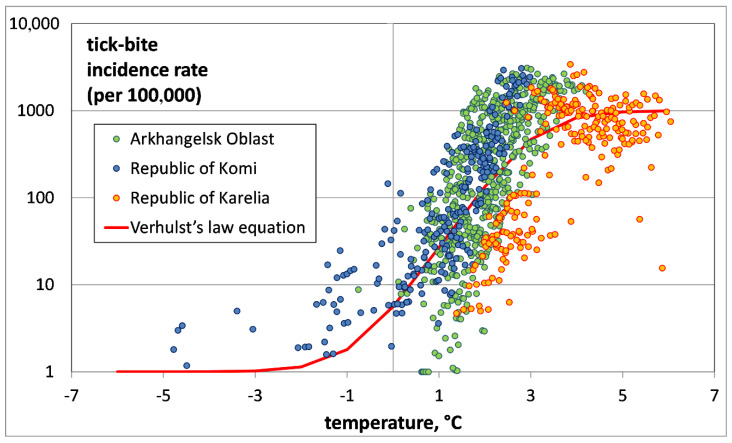
Regional distribution of tick-bite incidence rate versus average annual model air temperature for all administrative units of the Arkhangelsk Oblast, Karelia and Komi.

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
