# Peer review of "Study of the Relationship between the Average Annual Temperature of Atmospheric Air and the Number of Tick-Bitten Humans in the North of European Russia"

_ijerph, 2020, doi:10.3390/ijerph17218006_

Round 1
Reviewer 1 Report
My considerations are in the attached file.

Author Response
|
Page |
Text |
Reviewer note |
Reviewer suggestion |
Correction |
|
1 |
Ixodid ticks are known to be carriers of many pathogens |
Realce |
vectors |
Ixodid ticks are known to be vectors of many pathogens |
|
1 |
infectious diseases in humans. |
Realce |
such as .... please describe these pathogens. |
infectious such as tick-borne encephalitis and Lyme borreliosis, Human Monocytic Ehrlichiosis, Human Granulocytic Anaplasmosis and others |
|
1 |
Ixodid ticks are known to be carriers of many pathogens that cause dangerous infectious diseases in humans. Therefore, a quantitative estimate of tick population is necessary when developing preventive measures within any administrative territory. Usually the classic method of flagging, collection on a sheet, is applied. However, the results for separate individual areas where ticks were collected do not represent in a relevant manner the abundance of ixodidae over very large areas, such as the European North of Russia. Moreover, the monitoring of ticks abundance in Russia is very fragmentary [1]. |
Nota |
This entire paragraph presents classic information about ixodids and only one reference is cited. Could it be that this unique work really brought all this information in an unprecedented way? I suggest review. |
We cite of fundamental researches of I. persulcatus or Taiga tick at the end of the manuscript: 19. Korenberg, E.I., Jukova, V.I., Shatkauskus, A.V., Bushueva, L.K. Distribution of the taiga tick (Ixodes persulcatus) in the USSR [Russian]. J Zoology 1969, 48, 1003-1014. 23. Balashov, Yu.S. Ixodid ticks-parasites and vectors of diseases; Nauka: St. Petersburg, Russia, 1998; 287 p. [in Russian]. 24. Taiga Tick Ixodes persulcatus Schulze (Acarina, Ixodidae): Morphology, Systematics, Ecology, Medical Significance; ed. by Filippova N.A.; Nauka: Leningrad, USSR, 1985; 416 p [in Russian]. |
|
1 |
In our study ticks` abundance is estimated indirectly: we compare the number of tick victims over a number of years. |
Realce |
What did you compare your data with? This is confusing. |
In our research ticks` abundance is estimated indirectly: we study the number of tick victims over a number of years. |
|
2 |
varied during the study period, for the comparison purposes we used the tick-bite incidence rate (TBIR), that is the number of people bitten by ticks during the year, per 100 thousand of the local population. |
Realce |
This looks abstract and/or material and methods, not introduction. |
Transferred to Materials and Methods |
|
3 |
In total, 187,722 tick bites have been registered over the period under study: 102,791 in the Arkhangelsk Oblast; 18,690 in the Komi Republic and 66,241 in the Republic of Karelia. Since many years the population has been encouraged to seek medical assistance when having a tickbite and this is common procedure in Russia. |
Realce |
This is result, not MM |
Transferred to results |
|
4 |
|
Nota |
Please see: Bugmyrin, S.V., et al., Distribution of Ixodes ricinus and I. persulcatus ticks in southern Karelia (Russia). Ticks Tick-borne Dis. (2012), http://dx.doi.org/10.1016/j.ttbdis.2012.07.004 |
This research unlike our study use classic method of flagging. They also analyze Ixodes ricinus and I. persulcatus ticks only in the south area of Respublika Karelia. We try to understand the distribution of Ixodes ricinus and I. persulcatus over large territory and use method of human tick-bite incidence rate.
|
|
4 |
Average annual air temperatures range from negative in the north of the Komi Republic to +6 °С in the south of Karelia. |
Realce |
In my experience, it is very unlikely that a tick can survive in these conditions. Some species of the genus Haemaphysalis can maintain their biological cycle from 12ºC, but below that, I am particularly unaware. |
I. persulcatus survives in extremely hard temperature and precipitation conditions. Usually the temperature conditions for it described as: ” For the period with temperatures above +5°C, the accumulated temperatures of 1600–1900° constitute low heat provision for I. persulcatus”. We prefer to operate with annual mean temperature. The second name of I. persulcatus is taiga tick and it explains a lot about it ecology. Please find more information about I. persulcatus ecology: 1. Sirotkin, M & Korenberg, Eduard. (2018). Influence of Abiotic Factors on Different Developmental Stages of the Taiga Tick Ixodes persulcatus and the Sheep Tick Ixodes ricinus. Entomological Review. 98. 496-513. 10.1134/S0013873818040115. 2. Jaenson TG, Värv K, Fröjdman I, et al. First evidence of established populations of the taiga tick Ixodes persulcatus (Acari: Ixodidae) in Sweden. Parasit Vectors. 2016;9(1):377. Published 2016 Jul 1. doi:10.1186/s13071-016-1658-3 3. Igor Uspensky. The taiga tick Ixodes persulcatus (Acari: Ixodidae), the main vector of Borrelia burgdorferi sensu lato in Eurasia. In book: Lyme Disease (e-book), (pp.16 p.)Edition: SMGebook (www.smgebooks.com)Publisher: SM Group. https://www.researchgate.net/publication/ 307584073_The_taiga_tick_Ixodes_persulcatus_ Acari_Ixodidae_the_main_vector_of_Borrelia_ burgdorferi_sensu_lato_in_Eurasia
|
|
4 |
precipitation is 400-750 mm, and significant climate changes, especially |
Realce |
small... (??????) |
Due to very low temperatures such small precipitation volume forms humid climate condition when precipitation exceed evaporation and drain. Bogs and swamp forests are widely distributed in the area of our research. Please, see information about I. persulcatus ecology listed above. |
|
4 |
of Microsoft Excel interpolation tools |
Realce |
Why not use specific software to analyze this material? I believe that there are other more suitable software. |
Our primary tool to analyze spatial distribution of ticks and temperature is ArcGIS – geographical information system of professional level. Microsoft Excel is used as an additional software for simple calculations. |
|
5 |
accordingly, those places provide favorable habitat for ixodid ticks. |
Realce |
Too cold for the maintenance of tick biology. Why was not the analysis of the species of ticks that were identified in cases of tick-bite? |
Please, see our response for page 4. |
|
6 |
The local temperature varies between -7 °С and +3 °С. |
Realce |
Too cold for the maintenance of tick biology. |
Please, see our response for page 4. |
|
6 |
Analysis of the TBIR distribution as a function of air temperature shows that it does not follow the exponential “Malthus equation”, as indicated by its relatively low correlation coefficient of 0.67 (p <0.0001). |
Realce |
What statistical test was performed to obtain this significance value? What is the post-test? What software? |
We use Pearson correlation coefficient test for calculate linear relation between TBIR distribution and air temperature. As a post-test t-test was applied. We also use Microsoft Excel for this purpose. |
|
8 |
institutions, that recorded all medical care encounters in connection with tick bites. |
Realce |
What % of tick bites have resulted in disease? It is extremely important to consider this data as also. |
Tick-borne encephalitis prevalence data described in our earlier papers: Tokarevich, N., Tronin, A., Blinova, O., Buzinov, R., Boltenkov, V., Yurasova, E., Nurse, J. The impact of climate change on the expansion of Ixodes persulcatus habitat and the incidence of tick borne encephalitis in the north of European Russia. Global Health Action 2011, 4. 8448. doi: 10.3402/gha.v4i0.8448. Tokarevich, N., Tronin, A., Gnativ, B., Revich, B., Blinova, O. and Evengard, B. Impact of air temperature variation on the ixodid ticks habitat and tick-borne encephalitis incidence in the Russian Arctic: the case of the Komi Republic. International Journal of Circumpolar Health 2017, 76(1). 1298882. 10.1080/22423982.2017.1298882. |
|
8 |
be 1400–1500 °C over |
Realce |
What?????? |
It is the sum of air temperatures either to be 1400–1500 °C over the period with stable average daily temperatures exceeding 10 °C. It is accepted description of temperature conditions for I. persulcatus. The value 1400–1500 °C can confuse. Please find more information about I. persulcatus ecology: 4. Sirotkin, M & Korenberg, Eduard. (2018). Influence of Abiotic Factors on Different Developmental Stages of the Taiga Tick Ixodes persulcatus and the Sheep Tick Ixodes ricinus. Entomological Review. 98. 496-513. 10.1134/S0013873818040115. 5. Jaenson TG, Värv K, Fröjdman I, et al. First evidence of established populations of the taiga tick Ixodes persulcatus (Acari: Ixodidae) in Sweden. Parasit Vectors. 2016;9(1):377. Published 2016 Jul 1. doi:10.1186/s13071-016-1658-3 6. Igor Uspensky. The taiga tick Ixodes persulcatus (Acari: Ixodidae), the main vector of Borrelia burgdorferi sensu lato in Eurasia. In book: Lyme Disease (e-book), (pp.16 p.)Edition: SMGebook (www.smgebooks.com)Publisher: SM Group. https://www.researchgate.net/publication/ 307584073_The_taiga_tick_Ixodes_persulcatus_ Acari_Ixodidae_the_main_vector_of_Borrelia_ burgdorferi_sensu_lato_in_Eurasia |
Reviewer 2 Report
Review on the manuscript entitled "Study of the relationship between the average annual temperature of atmospheric..." by Tronin et al.
I have enjoyed reading the manuscript. It is ready for publication as it is. I do have only one comment:
Why the authors have not looked at the relationship between temperature and first or last incidence of tick bite of each year. It is possible with warming climate, those incidences shifted to earlier season in spring and or later season in autumn.
This comment can be ignored since the presented data are enough for a publication.
Author Response
|
Reviewer note |
Correction |
|
I do have only one comment: Why the authors have not looked at the relationship between temperature and first or last incidence of tick bite of each year. It is possible with warming climate, those incidences shifted to earlier season in spring and or later season in autumn. This comment can be ignored since the presented data are enough for a publication. |
Dear reviewer, thank you for your very valuable comment. 1) We really find such type of sign pointed on warming climate. We discover in our previous research, the length of tick bite reporting period in 2000-2009 expanded in comparison with 1980-1989 in 20-fold in the northern group, twofold in the central group, and 1.5-fold in the southern group of municipal districts of Arkhangelsk Oblast (one of study area in current research). [Tokarevich, N., Tronin, A., Blinova, O., Buzinov, R., Boltenkov, V., Yurasova, E., Nurse, J. The impact of climate change on the expansion of Ixodes persulcatus habitat and the incidence of tick borne encephalitis in the north of European Russia. Global Health Action 2011, 4. 8448. doi: 10.3402/gha.v4i0.8448] 2) This shift of vegetation period related with warming climate will be the central point of the next research where we will use satellite remote sensing data to detect learly/late vegetation in Arctic area. |
Reviewer 3 Report
The authors present analyzes on tick-bite incidence rate depending on temperature changes in the last decades in the North of European Russia. The epidemiological data from three districts located in Russian taiga and tundra was presented. These areas are representing the northern range of distribution of Ixodes ticks. The meteorological data of over 60 years was used to calculate the monthly average temperature in each area. In general, the study is interesting in its field and contains complex data. Nevertheless, I have some comments on the study, and I recommend the major revision before the publication.
General issues:
Please separate the assumptions from the results. Some parts of the results belong to the discussion. Please move them.
Please use “the North of European Russia” instead “European North of Russia” in the whole manuscript, figures, and title.
Please do not use the expression “tick victims” and use “tick-bitten humans/patients” in the title and elsewhere.
When writing the temperature, please delete a gap between the number and the symbol of degree, e.g. 5°C.
The part of the results about the tick bites over time is very scarce.
Please use sentences in the passive voice.
Please change the titles of the figures. They should be more descriptive so they are understandable without reading the text.
Please note that the changes in the tick-bite incidence rate over time can be a result of better health care not only increase of temperature in the last decades.
Tittle
Please change it according to the hints above.
Lines:
25, 27 and elsewhere: “tick abundance”
25: “In our study, the official statistics of the number of tick-bitten humans were used”
39: “Usually, the classic method of flagging the vegetation is applied”. Please add a reference describing this method.
44: “In our study, tick abundance was estimated indirectly: the number of tick-bitten humans over years was analyzed”.
46: please use the passive voice
50: “where the flagging method is not efficient due to the small size of the tick population”
52: “dynamics of tick risk in the population”
Figure 1: please add to this map a zoomed-out view of the location in Europe
65: please add a comma after “here”
80: “For many years”
83: “Figure 2 represents…”- this sentence and the figure belongs to the results. Please consider replacing it.
83,137,233,268: please use the full name of the abbreviation when you use it for the first time in a new chapter
85: please use a gap before “TBIR”
83,137,233,268: please use the full name of the abbreviation (TBIR) when you use it for the first time in a new chapter
91: “The studied territory is located in the North of European Russia”
96-97: Authors mentioned only mammalian fauna. What about other animals that could serve as tick hosts?
110, 121: Please provide the version of the programs, producer, and country
114: “here:… is the monthly…”
120: please use a passive voice
125: “mATj is the average annual model air temperature [°С]; ATj is the average annual observed air temperature [°С]; a, b, c are calculated coefficients”
128: “Hereinafter, all..”
130: “The mATJ and TBIR values were used for all…”
136: please change the expression “tick victims”
137-139: please use passive voice sentences
Figure 3: please improve the title and the description of the figure
148-149: please rewrite these sentences
150: “the first group of samples”
135: “The second group”
155: “… is the first group,.. to analyze the dependence of the number of tick bites on the temperature”
157: this assumption should be in the discussion
160-161: please use a passive voice. At the end of the sentence please put the reference to Fig. 4 and delete the last sentence in this paragraph.
Figure 4: as mentioned above. Please describe the zones and mentioned used colors
166-177: “Our analysis of the air temperature dependence of nonzero TBIRs has revealed three TBIR zones (Figure 4): 1 - the invasion zone (blue color), 2 - the exponential growth zone (red), 3 - the saturation zone (green)”. And then please delete the following parts: “and in figure 4 it is marked out in blue”, “and in Figure 4 it is marked out in red”, “In Figure 4 the zone is marked out in green”
The sentence from line 173 should be moved to line 172. It should not be in a new paragraph.
179-181: “Perhaps…” – this sentence should be part of the discussion
186: “As mentioned earlier, the sample…”
189-191: “here: TBIR is the number of tick-bitten humans per 100 thousand of the population; TBIR0 is the initial number of tick bites per 100 thousand of the population; K is the carrying capacity (maximum possible size of a population); r is the growth rate; mATj is the modeled air temperature”
192-195: please use passive voice sentences
200: “in the North of European Russia”
201-203: The last sentence belongs to discussion not results
204: “Dependence of TBIR on air temperature by regions and over time”
205-206: “The dependence of TBIR on temperatures in three regions: Arkhangelsk Oblast, Republic of Karelia, and the Komi Republic is presented in Figure 6”
210-217: the assumptions are part of the discussion
225: “the North of European Russia”. “The northward shift of the forest borders leads to the dispersal of many species of wild mammals which are…”
227: “tick habitat”
228: “distribution range”
230: please use a passive voice
238: “human exposure to ticks”
240: “the incidence of tick-borne encephalitis (TBE)”
245: “ increase the risk of exposure…”
253: please add a reference
265: please add a comma before “TBIR”
279: “the size of tick population” or the exposure to ticks remains constant
Author Response
|
Reviewer suggestion |
Correction |
|
General issues: |
|
|
Please separate the assumptions from the results. Some parts of the results belong to the discussion. Please move them. |
In some cases we have change the text, in some cases assumptions was moved to discussion. |
|
Please use “the North of European Russia” instead “European North of Russia” in the whole manuscript, figures, and title. |
All cases “the European North of Russia” changed to “the North of European Russia”. |
|
Please do not use the expression “tick victims” and use “tick-bitten humans/patients” in the title and elsewhere. |
All cases “tick victims” changed to “tick-bitten humans/patients”. |
|
When writing the temperature, please delete a gap between the number and the symbol of degree, e.g. 5°C. |
All spaces between the number and the symbol of degree are deleted. |
|
The part of the results about the tick bites over time is very scarce. |
All cases of tick-bit incidents registered by medical organizations were involved in processing. Time series analysis of the number of tick-bitten humans in the Arkhangelsk region and the Komi Republic was executed earlier in our researches: Tokarevich, N., Tronin, A., Blinova, O., Buzinov, R., Boltenkov, V., Yurasova, E., Nurse, J. The impact of climate change on the expansion of Ixodes persulcatus habitat and the incidence of tick borne encephalitis in the north of European Russia. Global Health Action 2011, 4. 8448. doi: 10.3402/gha.v4i0.8448. Tokarevich, N., Tronin, A., Gnativ, B., Revich, B., Blinova, O. and Evengard, B. Impact of air temperature variation on the ixodid ticks habitat and tick-borne encephalitis incidence in the Russian Arctic: the case of the Komi Republic. International Journal of Circumpolar Health 2017, 76(1). 1298882. 10.1080/22423982.2017.1298882.
|
|
Please use sentences in the passive voice. |
We have changed all marked sentences to passive voice. |
|
Please change the titles of the figures. They should be more descriptive so they are understandable without reading the text. |
Figure titles were changed – see below. |
|
Please note that the changes in the tick-bite incidence rate over time can be a result of better health care not only increase of temperature in the last decades. |
The number of tick-bitten humans in time depends on the level of medical care in the region, on population knowledge about ticks danger to the health and social conditions (see “Discussion”). The contribution of these factors and negligible in comparison with climate change from our point of view. We confirm our thesis with 1) tick expansion to the North, 2) sharp rise of tick-borne encephalitis cases, 3) growth of seroprevalence to tick-borne encephalitis among local population (Tokarevich N ,Kazakovtsev S, Stoyanova N, Gnativ R, Blinova O, Revich B. Seroprevalence of tick-borne diseases in the population of the European North of Russia.. 2017.v 6, I. 1.), 4) reduction of the number of tick-bitten humans in the Republic of Karelia last years. It can not be related with decreasing medical care level or people awareness about tick danger. |
|
Tittle Please change it according to the hints above. |
The title of the manuscript was changed. |
|
Lines: |
|
|
25, 27 and elsewhere: “tick abundance” |
“Ticks’ abundance” were replaced to “tick abundance” |
|
25: “In our study, the official statistics of the number of tick-bitten humans were used” |
Replaced to: “In our study, the official statistics of the number of tick-bitten humans were used” |
|
39: “Usually, the classic method of flagging the vegetation is applied”. Please add a reference describing this method. |
Taiga Tick Ixodes persulcatus Schulze (Acarina, Ixodidae): Morphology, Systematics, Ecology, Medical Significance; ed. by Filippova N.A.; Nauka: Leningrad, USSR, 1985; 416 p. |
|
44: “In our study, tick abundance was estimated indirectly: the number of tick-bitten humans over years was analyzed”. |
Replaced to: “In our study, tick abundance was estimated indirectly: the number of tick-bitten humans over years was analyzed” |
|
46: please use the passive voice |
Since the habitancy varied during the study period, for the comparison purposes the tick-bite incidence rate (TBIR) was used, that is the number of humans bitten by ticks during the year, per 100 thousand of the local population. |
|
50: “where the flagging method is not efficient due to the small size of the tick population” |
Replaced to: “where the flagging method is not efficient due to the small size of the tick population” |
|
52: “dynamics of tick risk in the population” |
Do not agree Replace to “In addition, long-period TBIR monitoring makes it possible to register the time dependence of tick abundance over the large territory” |
|
Figure 1: please add to this map a zoomed-out view of the location in Europe |
We have added a zoomed-out view of the location in Europe. |
|
65: please add a comma after “here” |
Set comma |
|
80: “For many years” |
“Since many years” replaced “For many years” |
|
83: “Figure 2 represents…”- this sentence and the figure belongs to the results. Please consider replacing it. |
Moved to Results |
|
83,137,233,268: please use the full name of the abbreviation when you use it for the first time in a new chapter |
All abbreviation (TBIR) changed to the full name for the first time in a new chapter |
|
85: please use a gap before “TBIR” |
Set space |
|
83,137,233,268: please use the full name of the abbreviation (TBIR) when you use it for the first time in a new chapter |
All abbreviation (TBIR) changed to the full name for the first time in a new chapter |
|
91: “The studied territory is located in the North of European Russia” |
“The territory studied lies in the North of the European Russia” replaced to “The studied territory is located in the North of European Russia” |
|
96-97: Authors mentioned only mammalian fauna. What about other animals that could serve as tick hosts? |
Wide set of migratory and sedentary bird species are very common, especially in the north part of research area – from Golden eagle to Little bunting and from Whooper swan to Eurasian bullfinch. A few Amphibian species like Common frog, Moor frog and Common toad could be found in this area. |
|
110, 121: Please provide the version of the programs, producer, and country |
Version of the programs, producer, and country is pointed. |
|
114: “here:… is the monthly…” |
Set article the |
|
120: please use a passive voice |
a second-order polynomial function was applied to the average |
|
125: “mATj is the average annual model air temperature [°С]; ATj is the average annual observed air temperature [°С]; a, b, c are calculated coefficients” |
“mATj is the average annual model air temperature, °С, ATj is average annual observed air temperature, °С, a, b, c are calculated coefficients” replaced to “mATj is the average annual model air temperature [°С]; ATj is the average annual observed air temperature [°С]; a, b, c are calculated coefficients” |
|
128: “Hereinafter, all..” |
“Hereinafter all” changed to “Hereinafter, all..” |
|
130: “The mATJ and TBIR values were used for all…” |
“Here we used of mATj and TBIR values for all” replaced to “The mATJ and TBIR values were used for all…” |
|
136: please change the expression “tick victims” |
“tick victims” replaced to “tick-bitten humans” |
|
137-139: please use passive voice sentences |
To study the relationship between mATj and TBIR a two-dimensional diagram was constructed: here the abscissa is mATj, and the ordinate is TBIR on a logarithmic scale (see Figure 3). To plot a chart on a logarithmic scale to base 10 for the TBIR values, "0" in our sample was replaced with "1". |
|
Figure 3: please improve the title and the description of the figure |
Figure 3. Tick-bite incidence rate (ordinate) versus average annual model air temperature (abscissa) for all administrative units of the Arkhangelsk Oblast, Karelia and Komi. TBIR on a logarithmic scale. Separate groups are highlighted in the figure. The description is given in the text. |
|
148-149: please rewrite these sentences |
Rewrite as “It is much more convenient to analyze samples in this form. In our opinion, this statistical population involves at least two groups:” |
|
150: “the first group of samples” |
“The first sample” to “the first group of samples” |
|
155: “The second group” |
“The second sample” to “The second group” |
|
155: “… is the first group,.. to analyze the dependence of the number of tick bites on the temperature” |
“to analyze the temperature dependence of the number of tick bites” to “ to analyze the dependence of the number of tick bites on the temperature” |
|
157: this assumption should be in the discussion |
Replace “Of greatest interest is the first sample, which makes it possible to analyze the temperature dependence of the number of tick bites and to get some new information about the ecology of ixodidae, i.e. the temperature limitations for Ixodes persulcatus distribution” to “Of greatest interest is the first group, which provides information about the dependence of the number of tick bites on the temperature.” |
|
160-161: please use a passive voice. At the end of the sentence please put the reference to Fig. 4 and delete the last sentence in this paragraph. |
For more detailed investigation of the first group, second group was deleted by removing all points corresponding to TBIRs between 0 and 1, i.e. negligibly small values (see Figure 4). |
|
Figure 4: as mentioned above. Please describe the zones and mentioned used colors |
Figure 4. Nonzero tick-bite incidence rate versus average annual model air temperature for all administrative units of the Arkhangelsk Oblast, Karelia and Komi. The highlighted areas are represented as the invasion zone (blue color), the exponential growth zone (red) and the saturation zone (green). |
|
166-177: “Our analysis of the air temperature dependence of nonzero TBIRs has revealed three TBIR zones (Figure 4): 1 - the invasion zone (blue color), 2 - the exponential growth zone (red), 3 - the saturation zone (green)”. And then please delete the following parts: “and in figure 4 it is marked out in blue”, “and in Figure 4 it is marked out in red”, “In Figure 4 the zone is marked out in green” |
“Our analysis of the air temperature dependence of non-zero TBIRs has revealed three TBIR zones in Figure 4: 1 - the invasion zone, 2 - the exponential growth zone, 3 - the saturation zone” replaced to “Our analysis of the air temperature dependence of nonzero TBIRs has revealed three TBIR zones (Figure 4): 1 - the invasion zone (blue color), 2 - the exponential growth zone (red), 3 - the saturation zone (green)”. |
|
The sentence from line 173 should be moved to line 172. It should not be in a new paragraph. |
new paragraph deleted |
|
179-181: “Perhaps…” – this sentence should be part of the discussion |
Remove the word “Perhaps…” |
|
186: “As mentioned earlier, the sample…” |
“Earlier it was mentioned already replaced” to “ As mentioned earlier, the sample…” |
|
189-191: “here: TBIR is the number of tick-bitten humans per 100 thousand of the population; TBIR0 is the initial number of tick bites per 100 thousand of the population; K is the carrying capacity (maximum possible size of a population); r is the growth rate; mATj is the modeled air temperature” |
“here: TBIR is the number of tick victims per 100 thousand of population, TBIR0 is initial number of tick victims per 100 thousand of population, K is the carrying capacity (maximum possible size of a population), r is the growth rate, mATj is the modeled air temperature.” Changed to “here: TBIR is the number of tick-bitten humans per 100 thousand of the population; TBIR0 is the initial number of tick bites per 100 thousand of the population; K is the carrying capacity (maximum possible size of a population); r is the growth rate; mATj is the modeled air temperature”
|
|
192-195: please use passive voice sentences |
To calculate those coefficients (r and TBIR0), the distribution for the Arkhangelsk Oblast was used, as the most representative distribution in the area of exponential growth. Based on the distribution (Figure 4) we took K for 1000. We added one to the calculated value for its correct representation on the graph. The graph of the “Verhulst equation“ is shown in Figure 5. |
|
200: “in the North of European Russia” |
“in the North of European Russia” |
|
201-203: The last sentence belongs to discussion not results |
Removed |
|
204: “Dependence of TBIR on air temperature by regions and over time” |
“Dependence of TBIR on air temperature by regions” to “Dependence of TBIR on air temperature by regions and over time” |
|
205-206: “The dependence of TBIR on temperatures in three regions: Arkhangelsk Oblast, Republic of Karelia, and the Komi Republic is presented in Figure 6” |
“Let us consider the distribution of TBIR versus temperature in three regions separately: Arkhangelsk Oblast, Republic of Karelia, and Komi Republic (Figure 6).” Replaced to “The dependence of TBIR on temperatures in three regions: Arkhangelsk Oblast, Republic of Karelia, and the Komi Republic is presented in Figure 6” |
|
210-217: the assumptions are part of the discussion |
Change “It can be assumed that the distribution in time is shifted from the region of low temperatures (Republic of Komi) to the region of high temperatures (Republic of Karelia)” to “It is observed the shift of the distribution in time from the region of low temperatures (Republic of Komi) to the region of high temperatures (Republic of Karelia).” And keep on same place. |
|
225: “the North of European Russia”. “The northward shift of the forest borders leads to the dispersal of many species of wild mammals which are…” |
“The northward shift of the forest limit implied northward dispersal of many species of wild mammals which are” to “The northward shift of the forest borders leads to the dispersal of many species of wild mammals which are” |
|
227: “tick habitat” |
“ticks’ habitat” to “tick habitat” |
|
230: please use a passive voice |
In our study, to estimate the dynamics of ticks population, long-term statistics provided by Rospotrebnadzor institutions, that recorded all medical care encounters in connection with tick bites was used |
|
238: “human exposure to ticks” |
“humans` exposure to ticks” to : “human exposure to ticks” |
|
240: “the incidence of tick-borne encephalitis (TBE)” |
“the incidence of tick-borne encephalitis” to “the incidence of tick-borne encephalitis (TBE)” |
|
245: “ increase the risk of exposure…” |
“increase the risks of exposure” To “ increase the risk of exposure…” |
|
253: please add a reference |
1. Balashov, Yu.S. Ixodid ticks-parasites and vectors of diseases; Nauka: St. Petersburg, Russia, 1998; 287 p. [in Russian]. 2. Taiga Tick Ixodes persulcatus Schulze (Acarina, Ixodidae): Morphology, Systematics, Ecology, Medical Significance; ed. by Filippova N.A.; Nauka: Leningrad, USSR, 1985; 416 p. [in Russian]. |
|
265: please add a comma before “TBIR” |
Set comma |
|
279 :“the size of tick population” or the exposure to ticks remains constant |
“the number of the ticks population” to :“the size of tick population” |
Round 2
Reviewer 1 Report
My suggestions and considerations - as well as those of the other reviewers - were met and substantially improved the manuscript.
All my doubts were also answered and when necessary, added to the body of the manuscript and I have no more suggestions and/or comments on the manuscript.
The language was also improved.
I am therefore in favorable position to accept the manuscript in question for further publication since the changes were very satisfactory and the manuscript is in accordance with the scope of IJERPH.
Reviewer 3 Report
The authors have made considerable changes to address my concerns and comments. I have only two slight remarks.
The Authors still use “ticks abundance” (e.g. L25, 27, 44, 257, 259, 271).
L41: "collection from a sheet"